# Hypertension as a sequela in patients of SARS-CoV-2 infection

**Ganxiao Chen**[1‡], **Xun Li**[2‡], **Zuojiong Gong**[2], **Hao Xia**[1], **Yao Wang**[2], **Xuefen Wang**[3], **Yan Huang**[1], **Hector Barajas-Martinez**[4], **Dan Hu**[1]*

**1** Department of Cardiology & Cardiovascular Research Institute, Renmin Hospital of Wuhan University, Wuhan, China, **2** Department of Infectious Diseases, Renmin Hospital of Wuhan University, Wuhan, China, **3** Nursing department, Renmin Hospital of Wuhan University, Wuhan, China, **4** Lankenau Institute for Medical Research, and Lankenau Heart Institute, Wynnewood, Pennsylvania and Jefferson Medical College, Philadelphia, Pennsylvania, United States of America

‡ These authors share first authorship on this work.
* hudan0716@hotmail.com, rm002646@whu.edu.cn

## Abstract

### Background

COVID-19 is a respiratory infectious disease caused by SARS-CoV-2, and cardiovascular damage is commonly observed in affected patients. We sought to investigate the effect of SARS-CoV-2 infection on cardiac injury and hypertension during the current coronavirus pandemic.

### Study design and methods

The clinical data of 366 hospitalized COVID-19-confirmed patients were analyzed. The clinical signs and laboratory findings were extracted from electronic medical records. Two independent, experienced clinicians reviewed and analyzed the data.

### Results

Cardiac injury was found in 11.19% (30/268) of enrolled patients. 93.33% (28/30) of cardiac injury cases were in the severe group. The laboratory findings indicated that white blood cells, neutrophils, procalcitonin, C-reactive protein, lactate, and lactic dehydrogenase were positively associated with cardiac injury marker. Compared with healthy controls, the 190 patients without prior hypertension have higher AngII level, of which 16 (8.42%) patients had a rise in blood pressure to the diagnostic criteria of hypertension during hospitalization, with a significantly increased level of the cTnI, procalcitonin, angiotensin-II (AngII) than those normal blood pressure ones. Multivariate analysis indicated that elevated age, cTnI, the history of hypertension, and diabetes were independent predictors for illness severity. The predictive model, based on the four parameters and gender, has a good ability to identify the clinical severity of COVID-19 in hospitalized patients (area under the curve: 0.932, sensitivity: 98.67%, specificity: 75.68%).

**Data Availability Statement:** Data cannot be shared publicly because of the contents including information that could compromise research participant privacy/consent. Data are available from the Renmin Hospital of Wuhan University Ethics

Committee (contact via whdxrmyy@126.com) for researchers who meet the criteria for access to confidential data.

**Funding:** The current work was supported by the National Natural Science Foundation Project of China (Grant No. 81670304, D.H.).

**Competing interests:** The authors report no relationships that could be construed as a conflict of interest.

**Abbreviations:** ACE2, angiotensin converting enzyme II; Ang II, Angiotensin II; AT1R, Angiotensin II type-1 receptor; COVID-19, coronavirus disease 2019; cTnI, cardiac troponin I; MERS, middle east respiratory syndrome; RAS, renin-angiotensin system; SARS, severe acute respiratory syndrome.

## Conclusion

Hypertension, sometimes accompanied by elevated cTnI, may occur in COVID-19 patients and become a sequela. Enhancing Ang II signaling, driven by SARS-CoV-2 infection, might play an important role in the renin-angiotensin system, and consequently lead to the development of hypertension in COVID-19.

## Introduction

In December 2019, an acute respiratory infectious disease known as "coronavirus disease 2019 (COVID-19)" caused by a novel coronavirus occurred in Wuhan, China [1, 2]. Whole-genome sequencing and systematic analysis showed that this novel. Coronavirus is a distinct clade from beta coronavirus associated with human severe acute respiratory syndrome (SARS) and Middle East respiratory syndrome (MERS) [3], and was now officially named "SARS-CoV-2" by World Health Organization. Both SARS-CoV and SARS-CoV-2 have been identified to use the angiotensin converting enzyme II (ACE2) receptor as the portal of entry into the affected cell [4, 5]. ACE2, a membrane-bound aminopeptidase, is highly expressed in the heart and lung [6, 7]. Although the main clinical features of COVID-19 are dominated by respiratory symptoms, many patients with severe cardiovascular damage have been reported by our team and others [8, 9]. Besides, patients with underlying cardiovascular diseases might have an increased risk of death [8]. So, understanding the damage to the cardiovascular system caused by SARS-CoV-2 and the underlying mechanisms is of great importance so that these patients can be treated timely, and the mortality can be reduced. In this retrospective cohort study, the clinical data of hospitalized COVID-19-confirmed patients were analyzed to explore the consequences of SARS-CoV-2 infection on the cardiovascular system.

## Materials and methods

### Study setting and population

There were 366 COVID-19-confirmed patients enrolled in this study, who were hospitalized in the Department of Infectious Diseases, Renmin Hospital of Wuhan University, from February 1 to May 1, 2020. Clinical severity was defined for all enrolled COVID-19 patients according to the guidelines of the National Health Commission of China, including four types as mild, moderate, severe, and critical types [10]. We divided the patients into the non-severe group (mild and moderate types) and the severe group (severe and critical type). Mild type is defined as mild clinical symptoms and no pneumonia manifestation found in imaging. Moderate cases refer to those who present with fever and respiratory tract symptoms, etc. And have pneumonia manifestations found in imaging. Patients considered severe had one of the following three conditions: respiratory distress and respiratory rate higher than 30 times per minute; fingertip blood oxygen saturation less than 93% at rest; partial arterial oxygen pressure (PaO2) / fraction of inspiration oxygen (FiO2) less than 300mmHg. Patients in critical type met one of the following criteria: respiratory failure, requiring mechanical ventilation; shock; multiple organ failure, requiring intensive care management. This study was reviewed and approved by the Medical Ethical Committee of Renmin Hospital of Wuhan University. All participants provided written informed consent and agreed to use their medical records for research purposes.

## Data collection

The clinical signs and laboratory findings were extracted from electronic medical records (Donghua Hospital Information System). Two independent, experienced clinicians reviewed and abstracted the data. The recorded information includes demographic data, potential comorbidities, symptoms, signs, laboratory test results. The serum level of hypersensitive troponin I (cTnI) exceeding >40 pg/mL was considered cardiac injury [11]. Blood pressures were obtained three fixed times in the morning using standard measurement. History of hypertension was defined as brachial blood pressure $\geq$ 140/90 mmHg or self-reported hypertension medication use before hospitalization. For patients without prior hypertension, elevated blood pressure was defined as blood pressure $\geq$ 140/90 mmHg more than 3 times during hospitalization.

## The processes of patient screening

The screening process for evaluating the effect of SARS-CoV-2 on the cardiovascular system is shown in **Fig 1**. Serum level of cardiac troponin I (cTnI) was tested in 276 of the 366 patients during hospitalization, among which 8 patients had a history of chronic heart disease (including ischemic heart disease, arrhythmia, valvular disease, and heart failure) and were therefore excluded. Thus, 268 patients were enrolled to evaluate the effect of SARS-CoV-2 on cardiac injury. Of the 366 patients, 278 had complete blood pressure data. Among these, 88 patients had a history of hypertension before hospitalization and were excluded; therefore, 190 patients were grouped to evaluate the effect of SARS-CoV-2 on blood pressure. Among all 366 subjects, 194 subjects had data available on serum level of cTnI and complete blood pressure data. After

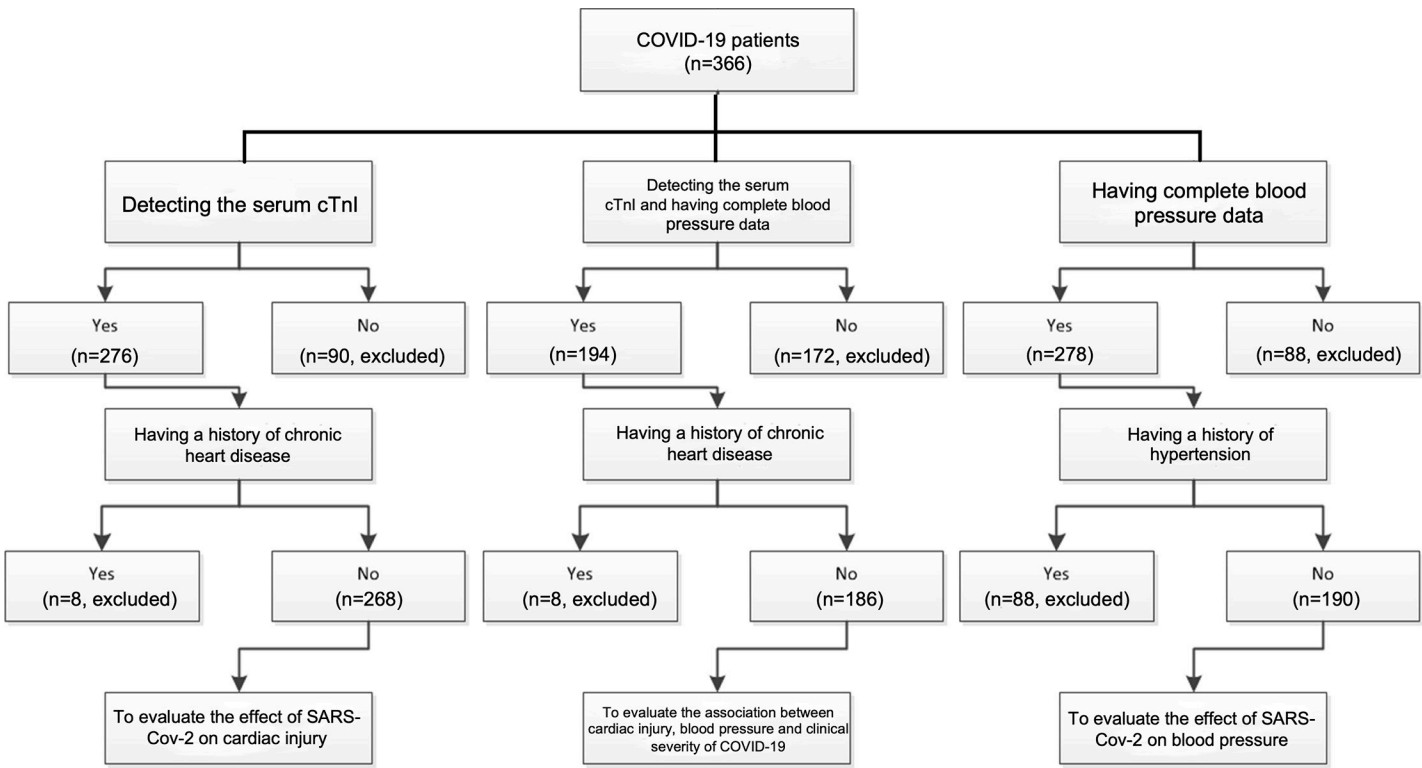

**Fig 1. The flow diagram of patient screening.** Chronic heart disease includes ischemic heart disease, arrhythmia, valvular disease, and heart failure.

the exclusion of the case with a history of chronic heart disease, 186 cases were included to evaluate the association between cTnI, blood pressure, and clinical severity of COVID-19.

## Statistical analysis

Student's *t-test* or the Mann-Whitney test was used to compare the mean of continuous variables, *Fisher's* exact test was used with limited data, the $\chi^2$ test was used to compare the proportion of categorical variables. Spearman correlation analysis was used to analyze the correlation between variables. The logistic regression model was used to determine factors associated with the clinical severity of COVID-19, and the analysis of receiver operating characteristic (ROC) curves was constructed according to standard procedures. The Youden index, defined as (sensitivity + specificity)– 1, was used to derive a reasonable cut-off value. Calibration of the risk prediction model, comparing the observed and predicted probability, was performed via a visual calibration plot in the R program. A P-value of < 0.05 was considered statistically significant. Statistical analysis was carried out using SPSS software version 21.0 and R version 3.0.

## Results

### The effect of SARS-CoV-2 on cardiac injury

The results showed that the median age of patients with or without cardiac injury was 74 y/o and 49 y/o with statistical significance (median [interquartile range]: 74 [73–86] vs. 49 [40–66] y/o, $p < 0.001$). Males were dominant in the cardiac injury group (86.67%). The cardiac injury was found in 11.19% (30/268) of patients, but 93.33% (28/30) of them were in the severe group. The proportion of cardiac injury was significantly lower in the non-severe group (1.75% vs. 18.18%, $p < 0.001$). Moreover, 66.67% (20/30) of cardiac injury patients in the severe group eventually died. The most frequent symptom of patients was fever, followed by cough, fatigue, dyspnea, and chest stuffiness. The incidences of cough, dyspnea, and chest stuffiness were significantly different between the patients with or without cardiac injury (93.33% vs. 68.91%, $p = 0.005$; 93.33% vs. 38.66%, $p < 0.001$; 86.67% vs. 36.97%, $p < 0.001$; respectively). Hypertension was the most frequent comorbidity, while the incidence of diabetes was significantly different between the two groups (33.33% vs. 14.29%, $p = 0.017$; **Table 1**). The laboratory findings indicated that the patients who suffered from cardiac injury had a higher level of white blood cells, neutrophils, monocytes, procalcitonin, C-reactive protein, lactate, and lactic dehydrogenase compared with the patients without cardiac injury (median [interquartile range]: 9.67 [5.62–13.73] vs. 5.93 [4.45–7.06] cells/L, $p < 0.001$; 5.52 [3.83–11.62] vs. 3.72 [2.95–5.47] cells/L, $p < 0.001$; 0.62 [0.42–0.76] vs. 0.44 [0.25–0.69] cells/L, $p = 0.037$; 630 [47.00–2750.00] vs. 60 [32.00–121.00] pg/mL, $p < 0.001$; 81.10 [14.20–142.80] vs. 41.40 [5.00–74.40] mg/L, $p < 0.001$; 2.10 [1.95–3.05] vs. 1.70 [1.15–2.00] mmol/L, p<0.001; 428 [325.00–765.00] vs. 275 [218.00–375.00] U/L, $p < 0.001$; respectively). Correlation analysis showed that white blood cells, neutrophils procalcitonin, C-reactive protein, lactate and lactic dehydrogenase were significantly associated with cTnI, the *r* values were 0.515 [95% CI, 0.394–0.632], 0.486 [95% CI, 0.358–0.591], 0.477 [95% CI, 0.352–0.581], 0.459 [95% CI, 0.338–0.566], 0.424 [95% CI, 0.273–0.559] and 0.438 [95% CI, 0.291–0.561], respectively (**Table 2**).

### The effect of SARS-CoV-2 on blood pressure

Of the 190 qualified patients, 16 (8.42%) patients had a rise in blood pressure during hospitalization, among which 6 patients were male, and 10 patients were female. As shown in **Table 3**, no significant differences were found when comparing the baseline demographics, including

**Table 1. Clinical characteristics of COVID-19 patients with or without cardiac injury.**

| | Total | Cardiac injury | |
|---|---|---|---|
| | (n = 268) | Non-injury (n = 238) | Injury (n = 30) |
| **Age (y/o), Median (IQR)** | 53 (42–69) | 49 (40–66) | 74 (73–86)*§ |
| **Gender (n, %)** | | | |
| Male | 144 (53.7) | 118 (49.58) | 26 (86.67)*£ |
| Female | 124 (46.3) | 120 (50.42) | 4 (13.33)*£ |
| **Clinical categories (n, %)** | | | |
| Non-severe | 114 (42.54) | 112 (47.06) | 2 (6.67)*£ |
| Severe | 154 (57.46) | 126 (52.94) | 28 (93.33)*£ |
| **Symptoms (n, %)** | | | |
| Fever | 224 (83.58) | 196 (82.35) | 28 (93.33)£ |
| Cough | 192 (71.64) | 164 (68.91) | 28 (93.33)*£ |
| Dyspnea | 120 (44.78) | 92 (38.66) | 28 (93.33)*£ |
| Chest stuffiness | 114 (42.54) | 88 (36.97) | 26 (86.67)*£ |
| Fatigue | 148 (55.22) | 130 (54.62) | 18 (60.00)£ |
| Muscle soreness | 54 (20.15) | 48 (20.17) | 6 (20.00)£ |
| **Comorbidities (n, %)** | | | |
| Hypertension | 94 (35.07) | 80 (33.61) | 14 (46.67)£ |
| Diabetes | 44 (16.42) | 34 (14.29) | 10 (33.33)*# |
| Chronic lung diseases | 10 (3.73) | 8 (3.36) | 2 (6.67)# |
| Chronic kidney diseases | 4 (1.49) | 2 (0.84) | 2 (6.67)¶ |
| Gastrointestinal diseases | 2 (0.75) | 2 (0.84) | 0 (0) |
| Malignant tumor | 4(1.49) | 0 (0) | 4 (13.33) |

Student's *t-test*, *χ2 test*, and *Fisher's exact tests* were used to compare the age, gender, clinical category, symptoms, and comorbidities between the two groups (§ Two-Sample T-test, £ Pearson's chi-square test, # continuous correction Chi-square test, ¶ Fisher's exact test).
*P < 0.05 is considered statistically significant.

age, gender, clinical category, symptoms, and comorbidities between patients with or without elevated blood pressure. Compared with the patients without elevated blood pressure, the level of cTnI and procalcitonin in the 16 patients rose significantly (median [interquartile range]:

**Table 2. The laboratory findings of COVID-19 patients with or without cardiac injury.**

| Laboratory findings | Non-cardiac injury (median, IQR) | Cardiac injury (median, IQR) | Normal range | *r* values | *P*-values |
|---|---|---|---|---|---|
| cTnI (pg/mL) | 7.5 (6.00–16.50) | 162 (68.20–757.50)* | 0–40 | 1.000 | < 0.001 |
| White blood cells (×10⁹ cells/L) | 5.93 (4.45–7.06) | 9.67 (5.62–13.73)* | 3.5–9.5 | 0.515 | < 0.001 |
| Neutrophils (×10⁹ cells/L) | 3.72 (2.95–5.47) | 5.52 (3.83–11.62)* | 1.8–6.3 | 0.486 | < 0.001 |
| Lymphocytes (×10⁹ cells/L) | 1.01 (0.59–1.25) | 0.67 (0.48–1.39)* | 1.1–3.2 | -0.230 | 0.001 |
| Monocytes (×10⁹ cells/L) | 0.44 (0.25–0.69) | 0.62 (0.42–0.76)* | 0.1–0.6 | 0.127 | 0.080 |
| Procalcitonin (pg/mL) | 60 (32.00–121.00) | 630 (47.00–2750.00)* | 0–100 | 0.477 | < 0.001 |
| C-reactive protein (mg/L) | 41.40 (5.00–74.40) | 81.10 (14.20–142.80)* | 0–10 | 0.459 | < 0.001 |
| Lactate (mmol/L) | 1.70 (1.15–2.00) | 2.10 (1.95–3.05)* | 0.5–1.5 | 0.424 | < 0.001 |
| Lactic dehydrogenase (U/L) | 275 (218.00–375.00) | 428 (325.00–765.00)* | 120–250 | 0.438 | < 0.001 |

The Mann-Whitney test was used to compare the differences between non-cardiac injury and cardiac injury groups
*P <0.05 is considered statistically significant. Spearman correlation analysis was used to analyze the correlation between the cTnI and other laboratory findings.

**Table 3. Clinical characteristics of COVID-19 patients without prior hypertension.**

| | Total | Blood pressure | |
|---|---|---|---|
| | (n = 190) | Normal | Elevated |
| | | (n = 174) | (n = 16) |
| **Age (y/o), Median (IQR)** | 54 (39–63) | 53 (39–63) | 60 (42–70)[§] |
| **Gender (n, %)** | | | |
| Male | 78 (41.05) | 72 (41.38) | 6 (37.50)[£] |
| Female | 112 (58.95) | 102 (58.62) | 10 (62.50)[£] |
| **Clinical categories (n, %)** | | | |
| Non-severe | 116 (61.05) | 108 (62.07) | 8 (50.00)[£] |
| Severe | 74 (38.95) | 66 (37.93) | 8 (50.00)[£] |
| **Symptoms (n, %)** | | | |
| Fever | 156 (82.11) | 142 (81.61) | 14 (87.50)[#] |
| Cough | 132 (69.47) | 120 (68.97) | 12 (75.00)[#] |
| Dyspnea | 70 (36.84) | 64 (36.78) | 6 (37.50)[£] |
| Chest stuffiness | 68 (35.79) | 62 (35.63) | 6 (37.50)[£] |
| Fatigue | 96 (50.53) | 88 (50.57) | 8 (50.00)[£] |
| Muscle soreness | 36 (18.95) | 34 (19.54) | 2 (12.50)[#] |
| **Comorbidities (n, %)** | | | |
| Diabetes | 28 (14.74) | 26 (14.94) | 2 (12.50)[#] |
| Chronic lung diseases | 6 (3.16) | 6 (3.45) | 0 (0) |
| Gastrointestinal diseases | 4 (2.11) | 4 (2.30) | 0 (0) |
| Thyroid disease | 2 (1.05) | 2 (1.15) | 0 (0) |
| Prostate disease | 2 (1.05) | 2 (1.15) | 0 (0) |

Student's *t-test*, $\chi 2$ *test*, and *Fisher's exact tests* were used to compare the age, gender, clinical category, symptoms, and comorbidities between the two groups ([§] Two-Sample T-test, [£] Pearson's chi-square test, [#] continuous correction Chi-square test). No significant differences were found.

22.00 [18.20–30.00] vs. 3.86 [2.49–5.15], $p < 0.001$; 82 [53–430] vs. 49 [28–73], $p = 0.023$; **Table 4**). Elevated systolic blood pressure was observed in most of the patients, while diastolic blood pressure was in the normal range. The median values of blood pressure and plasma cTnI levels changes of the 16 patients are shown in **Fig 2A**. Systolic blood pressure and cTnI

**Table 4. The laboratory findings of patients with or without elevated blood pressure.**

| Laboratory findings | Normal blood pressure | Elevated blood pressure | Normal range |
|---|---|---|---|
| | (median, IQR) | (median, IQR) | |
| cTnI (pg/mL) | 3.86 (2.49–5.15) | 22.00 (18.20–30.00)* | 0–40.00 |
| White blood cells (×10⁹cells/L) | 5.24 (3.87–7.00) | 4.86 (3.96–6.60) | 3.50–9.50 |
| Neutrophils (×10⁹cells/L) | 3.14 (2.48–4.96) | 3.64 (3.06–4.94) | 1.80–6.30 |
| Lymphocytes (×10⁹cells/L) | 1.13 (0.76–1.56) | 0.93 (0.57–1.23)* | 1.10–3.20 |
| Monocytes (×10⁹cells/L) | 0.44 (0.33–0.56) | 0.55 (0.25–0.61) | 0.10–0.60 |
| Hemoglobin (g/L) | 128 (118–136) | 118(107–137) | 115–150 |
| Procalcitonin (pg/mL) | 49 (28–73) | 82 (53–430)* | 0–100 |
| C-reactive protein (mg/L) | 20.00 (2.40–44.50) | 7.80 (3.51–33.20) | 0–10.00 |
| Lactic dehydrogenase (U/L) | 259 (206–313) | 259 (208–289) | 120–250 |

The Mann-Whitney test was used to compare the differences between non-hypertension and hypertension groups

*$P <0.05$ is considered statistically significant.

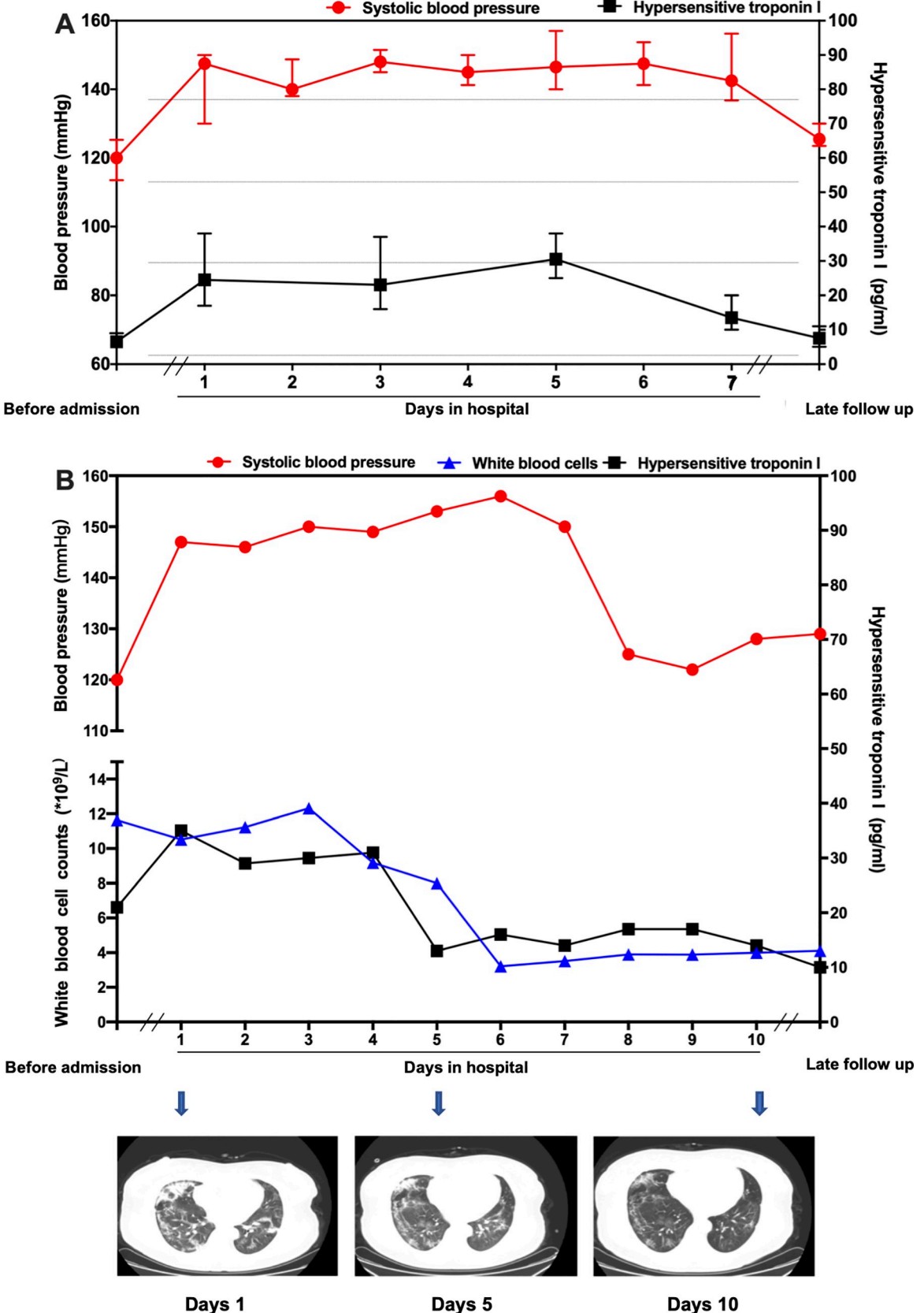

**Fig 2.** A. The systolic blood pressure and cTnI change of patients with elevated blood pressure. B. The systolic blood pressure, cTnI, white blood cells, and chest computed tomography changes of one patient with elevated blood pressure. Late follow-up: The 4[th] week after discharge from the hospital.

levels had a similar trend with the treatment time. In addition, the blood pressure, cTnI, and white blood cells were continuously monitored in one index case (**Fig 2B**). With effective treatment, the patient's condition improved with regards to symptoms and as evident on chest CT. Meanwhile, the systolic blood pressure and white blood cells reverted to the normal range, and the concentrations of cTnI were also gradually decreased. Among the 190 patients without prior hypertension, the serum levels of components in the renin-angiotensin system, including adrenocorticotrophic hormone, renin, angiotensin II (Ang II), and aldosterone, were detected in 28 patients. Comparison with healthy controls showed that AngII were significantly elevated in both the normal and elevated blood pressure groups (median [interquartile range]: 137.12 [123.63–161.67] vs. 87.90 [48.23–107.39] pg/mL, $p < 0.001$; 169.25 [142.17–186.98] vs. 87.90 [48.23–107.39] pg/mL, $p < 0.001$; respectively; **Table 5**). When comparing the RAS between the normal and elevated blood pressure groups, Ang II levels were significantly higher in the latter group (median [interquartile range]: 169.25 [142.17–186.98] vs. 137.12 [123.63–161.67] pg/mL, $p = 0.020$; **Table 6**).

## The association between cardiac injury, blood pressure, and clinical severity of COVID-19

Results of demographic and laboratory findings between the severe and non-severe group, based on guidelines of the National Health Commission of China, are shown in **Table 7**. Patients in the severe group were significantly older, with a greater proportion of males (median [interquartile range]: 66 [57–76] vs. 42 [33–51] y/o, $p < 0.001$; 60.00% vs. 40.54%, $p = 0.009$; respectively). In addition, the cTnI, white blood cells, neutrophils, procalcitonin, C-reactive protein, and lactic dehydrogenase of the severe group were significantly higher than those of the non-severe group (median [interquartile range]: 7.00 [5.78–27.00] vs. 5.68 [4.62–6.45] pg/mL, $p < 0.001$; 5.98 [4.60–10.00] vs. 5.07 [3.65–6.00] $\times 10^9$cells/mL, $p = 0.004$; 3.85 [3.02–8.27] vs. 2.67 [2.15–3.92] $\times 10^9$ cells/mL, $p < 0.001$; 67.50 [32.00–288.00] vs. 36.00 [23.00–57.00] pg/mL, $p = 0.005$; 48.55 [7.40–81.50] vs. 6.20 [0.50–28.00] pg/mL, $p < 0.001$; 316.00 [235.00–454.00] vs. 233.50 [187.50–292.00], $p < 0.001$; respectively). In contrast, patients in the severe group had a significantly lower level of lymphocytes median [interquartile range]: 0.78 [0.48–1.29] vs. 1.27 [0.96–1.73], $\times 10^9$ cells/mL, $p < 0.001$). Further univariate analysis revealed that the age, sex, cTnI, white blood cells, neutrophils, lymphocytes, C-reactive

**Table 5. The renin-angiotensin system in subjects with and without COVID-19.**

| Laboratory findings | Healthy control | Normal blood pressure | Elevated blood pressure |
| --- | --- | --- | --- |
| | (median, IQR) | (median, IQR) | (median, IQR) |
| Adrenocorticotrophic hormone (pg/mL) | 31.92 (21.6–39.67) | 27.55 (21.12–39.56)[§] | 33.03 (19.42–40.89)[§] |
| Renin (pg/mL) | 5.31 (3.21–8.75) | 6.35 (3.34–7.83)[†] | 5.86 (4.41–6.76)[†] |
| Angiotensin II (pg/mL) | 87.90 (48.23–107.39) | 137.12. (123.63–161.67)[*†] | 169.25 (142.17–186.98)[*†] |
| Aldosterone (pg/mL) | 150.05 (129.32–164.32) | 159.62 (119.16–169.30)[†] | 141.54 (118.63–154.83)[†] |

The normal and elevated blood pressure groups were compared with the healthy control group, respectively, by Student's $t$ test or the Mann-Whitney test ([§] Two-Sample T-test, [†] Mann Whitney U test/Wilcoxon Sum Rank test).

[*]$P < 0.05$ is considered statistically significant.

**Table 6. The renin-angiotensin system in subjects with and without elevated blood pressure.**

| Laboratory findings | Total | Normal blood pressure | Elevated blood pressure |
|---|---|---|---|
| | (median, IQR) | (median, IQR) | (median, IQR) |
| Adrenocorticotrophic hormone (pg/mL) | 30.56 (20.99–40.54) | 27.55 (21.12–39.56) | 33.03 (19.42–40.89) |
| Renin (pg/mL) | 6.01 (4.15–7.46) | 6.35 (3.34–7.83) | 5.86 (4.41–6.76) |
| Angiotensin α (pg/mL) | 153.67 (129.17–175.98) | 137.12. (123.63–161.67) | 169.25 (142.17–186.98)* |
| Aldosterone (pg/mL) | 142.11 (119.16–163.28) | 159.62 (119.16–169.30) | 141.54 (118.63–154.83) |

Student's *t* test was used to compare the differences between normal and elevated blood pressure groups

*$P < 0.05$ is considered statistically significant.

protein, lactic dehydrogenase, and history of hypertension and diabetes were significantly associated with the clinical severity of COVID-19. In the multivariate analysis, the age, cTnI and history of hypertension and diabetes remained significant independent predictors (OR = 1.11, 95% CI: 1.07–1.16, $p < 0.001$; OR = 1.08, 95% CI: 1.01–1.15, $p = 0.018$; OR = 7.19, 95% CI: 2.55–20.31, $p < 0.001$; OR = 4.28, 95% CI: 1.41–12.97, $p = 0.010$; **Table 8**). The receiver operating characteristic curve of the four independent predictors and gender for clinical severity of COVID-19 is shown in **Fig 3A** (AUC: 0.932, sensitivity: 98.67%, specificity: 75.68%). The calibration indicated that the model was well-calibrated (**Fig 3B**).

**Table 7. Clinical characteristics of severe and non-severe COVID-19 patients.**

| | Clinical Severity | | Normal range |
|---|---|---|---|
| | Non-severe | Severe | |
| | (n = 111, 59.68%) | (n = 75, 40.32%) | |
| **Age (y/o), Median (IQR)** | 42 (33–51) | 66 (57–76)*§ | - |
| **Gender (n, %)** | | | |
| Male | 45 (40.54) | 45 (60.00)*£ | - |
| Female | 66 (59.46) | 30 (40.00)*£ | - |
| **Clinical categories (n, %)** | | | |
| Elevated blood pressure | 6 (5.41) | 10 (13.33)£ | - |
| Hypertension | 15 (13.51) | 20 (26.67)*£ | - |
| Diabetes | 7 (6.31) | 21 (28.00)*£ | - |
| Chronic lung diseases | 7 (6.31) | 10 (13.33)£ | - |
| Chronic kidney diseases | 2 (1.8) | 2 (2.67)# | - |
| Gastrointestinal diseases | 2 (1.8) | 1 (1.33)# | - |
| Malignant tumor | 2 (1.8) | 0 (0) | - |
| **Laboratory findings** | | | |
| cTnI (pg/mL) | 5.68 (4.62–6.45) | 7.00 (5.78–27.00)*† | 0–40.00 |
| White blood cells (×10⁹cells/L) | 5.07 (3.65–6.00) | 5.98 (4.60–10.00)*† | 3.50–9.50 |
| Neutrophils (×10⁹cells/L) | 2.67 (2.15–3.92) | 3.85 (3.02–8.27)*† | 1.80–6.30 |
| Lymphocytes (×10⁹cells/L) | 1.27 (0.96–1.73) | 0.78 (0.48–1.29)*† | 1.10–3.20 |
| Monocytes (×10⁹cells/L) | 0.42 (0.34–0.54) | 0.44 (0.29–0.69)† | 0.10–0.60 |
| Procalcitonin (pg/mL) | 36.00 (23.00–57.00) | 67.50 (32.00–288.00)*† | 0–100.00 |
| C-reactive protein (mg/L) | 6.20 (0.50–28.00) | 48.55 (7.40–81.50)*† | 0–10.00 |
| Lactic dehydrogenase (U/L) | 233.50 (187.50–292.00) | 316.00 (235.00–454.00)*† | 120–250 |

Student's *t* test, Mann-Whitney test, *χ2 test* and *Fisher's exact tests* were used to compare the age, gender, and clinical category between the two groups (§ Two-Sample T-test, † Mann Whitney U test/Wilcoxon Sum Rank test, £ Pearson's chi-square test, # continuous correction Chi-square test).

*$P < 0.05$ is considered statistically significant.

**Table 8. Univariate and multivariate analysis for clinical severity of COVID-19.**

| | Odds ratio | 95% CI | P-value |
|---|---|---|---|
| **Univariate analysis** | | | |
| Age (years) | 1.12 | 1.09–1.15 | <0.001* |
| Male (%) | 2.20 | 1.21–4.00 | 0.010* |
| cTnI (pg/mL) | 1.13 | 1.05 - 1.22 | 0.002* |
| White blood cells (×10⁹cells/L) | 1.39 | 1.13 - 1.70 | 0.002* |
| Neutrophils (×10⁹cells/L) | 1.50 | 1.19–1.90 | 0.001* |
| Lymphocytes (×10⁹cells/L) | 0.23 | 0.10–0.54 | 0.001* |
| Procalcitonin (pg/mL) | 1.01 | 1.00–1.01 | 0.069 |
| C-reactive protein (mg/L) | 1.02 | 1.01–1.04 | 0.002* |
| Lactic dehydrogenase (U/L) | 1.01 | 1.00–1.01 | 0.001* |
| Hypertension (%) | 2.489 | 1.185–5.226 | 0.016* |
| Diabetes (%) | 5.78 | 2.31–14.45 | <0.001* |
| **Multivariate analysis** | | | |
| Age (years) | 1.11 | 1.07–1.16 | <0.001* |
| Male (%) | 1.38 | 0.57–3.37 | 0.479 |
| CTnI (pg/mL) | 1.08 | 1.01–1.15 | 0.018* |
| Hypertension (%) | 7.19 | 2.55–20.31 | <0.001* |
| Diabetes (%) | 4.28 | 1.41–12.97 | 0.010* |

The logistic regression model was used to determine factors associated with the clinical severity of COVID-19 according to Table 7

*P <0.05 is considered statistically significant.

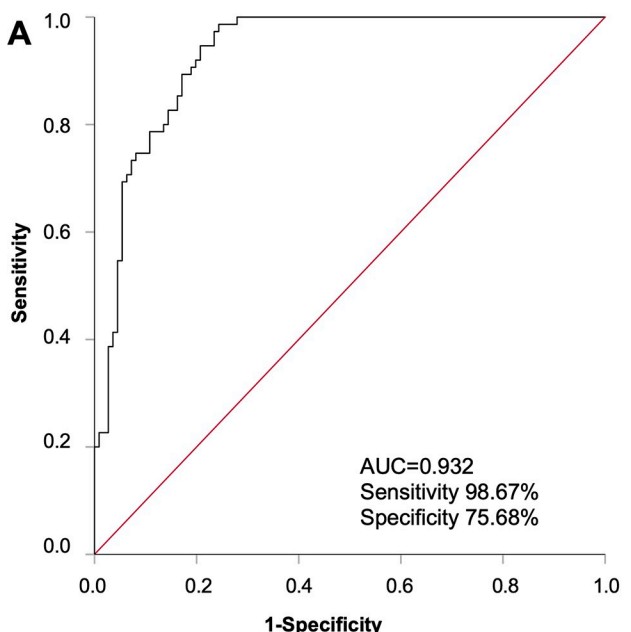
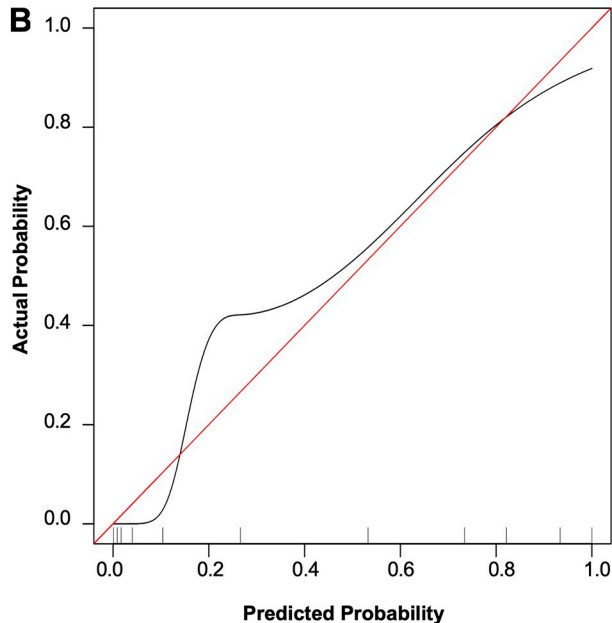

**Fig 3.** A. ROC curves of the age, cTnI, gender, and the presence of hypertension and diabetes for the identification of the severity of COVID-19. B. The calibration plot for the comparison of the predicted and actual probability. The X-axis and Y-axis represent the model-predicted and actual probability of MAE, respectively. The red line: perfect prediction. The black line: predictive performance of the model after bootstrapping (B = 1000 repetitions).

## Discussion

SARS-CoV-2 has been identified as a single-stranded enveloped 39 RNA virus belonging to the beta-coronavirus genus of the coronaviridae family [3]. These coronaviruses have a three-dimensional spike protein structure protein, which can closely bind the human ACE2 receptor. Therefore, the cells with ACE2 expression may act as target cells and be susceptible to SARS-CoV-2 infection [12]. ACE2 is a membrane-bound aminopeptidase with a vital role in the cardiovascular system [13, 14]. It is, therefore, reasonable to speculate that SARS-CoV-2 will act on the heart and blood vessels, with resultant changes in the cardiovascular system.

Serum cardiac troponin assays have been proposed as the recommended marker of cardiac injury in COVID-19 patients [15]. Huang and colleagues find that cTnI is increased substantially in 12.20% (5/41) Wuhan COVID-19 patients, in whom the diagnosis of the virus-related cardiac injury is made [8]. Another previous research also reported that patients with cardiac injury had higher levels of leukocyte counts, C-reactive protein, procalcitonin [16]. In this retrospective study, older patients with diabetes are more likely to suffer from cardiac injury. Our further analysis shows that the level of white blood cells, neutrophils, procalcitonin, C-reactive protein, lactate, and lactic dehydrogenase were positively associated with cardiac injury. Besides, the cardiac injury occurred mostly in severe patients. Consequently, we hypothesize that the severe type of COVID-19 characterized by acute inflammation response might be more prone to cardiac injury, especially in patients with preexisting cardiovascular disease. Chronic myocardial injury, acute nonischemic injury, and acute myocardial infarction have all been proposed as causes of cTnI elevation in COVID-19 patients [17]. One case study suggested that direct myopericardial involvement may be a complication of COVID-19 infection [18].

Understanding of the pathogenesis and complications of COVID-19 is still limited. Due to the lack of viral load quantification results in the literature, it is unclear whether the cardiac injury is directly related to viral load. The recent literature review has shown that although cTnI concentration is only marginally increased in all patients with COVID-19, (values above the 99[th] percentile upper reference limit in only 8–12% of positive patients), they are significantly increased in patients with severe disease [19]. Our study further suggests that cardiac injury is an independent risk factor for severe COVID-19 and in combination with age and other statistically significant comorbidities can be used to construct a logistic regression model of COVID-19 severity in hospitalized patients.

Sixteen patients without prior hypertension had a rise in blood pressure during hospitalization, and higher systolic blood pressure was observed in most of the patients. Except for lymphocytes and procalcitonin, no significant differences are found in patients with and without elevated blood pressure. This suggests that abnormal blood pressure may be caused independently of the inflammatory response. The RAS plays a critical role in the cardiovascular system, which includes a classical RAS axis (ACE-Ang II-AT1R pathway) and a non-classical RAS axis (ACE2-Ang 1-7-MasR-based pathway), counter-balancing role of the two axes regulates cardiovascular physiology and disease [20, 21]. ACE2 cleaves Ang II into the Ang 1–7, thus limiting substrate availability in the adverse ACE/Ang II/AT1 receptor axis [22, 23]. Keidar and colleagues found that ACE2 antihypertensive properties may be due to the degradation of angiotensin II [24]. In this study, the laboratory results of RAS show that Ang II level is elevated in the majority of patients without prior hypertension. Compared with normal blood pressure and healthy control groups, Ang II levels were significantly higher in elevated blood pressure groups. A possible mechanism may be the binding of SARS-CoV-2 to ACE2 thereby inhibiting degradation of angiotensin II leading to elevated blood pressure. Another hypothesis is that over activation of the RAS system promotes inflammatory response and cytokine storm, which stimulates the NADH/NADPH oxidase system and triggers cell contraction and

vasoconstriction, which then leads to COVID-19 related lung injury. Though the underlying mechanism remains to be elucidated, it is becoming evident that RAS plays a major role in hypertension and COVID-19 infection, as observed in our study. It has been noticed that recombinant human ACE2 is considered as a treatment for patients with COVID-19 (*ClinicalTrials.gov ID*: *NCT04287686*). This finding probably shades important implications for future treatment strategies. A recent long-term observational follow-up study of patients with COVID-19 reported nearly one-eighth of patients without previous renal dysfunction developed a reduction in glomerular filtration rate at follow-up. In addition, COVID-19 survivors suffer from relatively higher levels of depression, anxiety, and somatic symptoms (including fatigue or muscle weakness). Severe cases are more susceptible to the development of reduced pulmonary diffusion capacities [25]. Multiple above factors are capable of inducing hypertension in nonhypertensive patients. In addition, the median ages of these patients were 66.5 y/o. It seems that SARS-CoV-2 infection is just a trigger, and age plays a more important role.

On the other hand, sixteen patients with elevated blood pressure show significantly higher levels of cTnI than those normal blood pressure patients. Several studies have demonstrated Ang II direct or indirect effects on cardiomyocytes, some of which were related to pro-inflammatory and pro-hypertrophic responses [26]. Especially when the balance between the ACE and ACE2 was disrupted in COVID-19 patients, the increase in Ang II actions could lead to myocardial inflammation, oxidative stress, and myocyte apoptosis. This hypothesis explains why elevated blood pressure could occur in parallel with mild cardiac injury of COVID-19 patients.

## Study strength and limitations

In the present study, we propose that hypertension is probably a sequela of SARS-CoV-2 infection. Although a number of studies of COVID-19 have been reported, there are few reports about the sequela of the disease likely due to lack of long-term clinical follow-up, which also applies to our present research. Next, it is difficult to analyze whether the blood pressure of COVID-19 patients with preexisting hypertension is further increased. Consequently, many patients could not be incorporated in the analysis because of the history of hypertension, which results in a relatively low sample size. Besides, the present study uncovered rising Ang II as one possible mechanism that might result in hypertension in COVID-19. However, due to a lack of detection about ACE2 levels and other components, therefore, we cannot gain a comprehensive view of virus-induced imbalance of the RAS pathway.

## Conclusion

In summary, SARS-CoV-2 may impair cardiomyocytes by systemic acute inflammation response, and the cTnI is correlated with the severity of the infection. Accompanied by mild elevation in cTnI, spontaneous hypertension may occur in patients during hospitalization, and could become a sequela of SARS-CoV-2 infection, which may be associated with markedly elevated Ang II levels.

## Acknowledgments

We gratefully acknowledge contributions from all enrolled patients and related medical staff.

## Author Contributions

**Conceptualization:** Ganxiao Chen, Xun Li, Zuojiong Gong, Hao Xia, Yao Wang, Yan Huang, Hector Barajas-Martinez, Dan Hu.

**Data curation:** Ganxiao Chen, Xun Li, Zuojiong Gong, Yao Wang, Yan Huang, Dan Hu.

**Formal analysis:** Ganxiao Chen, Xun Li, Zuojiong Gong, Hao Xia, Xuefen Wang, Yan Huang.

**Funding acquisition:** Ganxiao Chen, Xun Li, Xuefen Wang.

**Investigation:** Ganxiao Chen, Xun Li.

**Methodology:** Ganxiao Chen, Xun Li, Hao Xia.

**Project administration:** Ganxiao Chen, Xun Li, Yao Wang, Xuefen Wang, Yan Huang, Hector Barajas-Martinez.

**Resources:** Ganxiao Chen, Hao Xia, Yao Wang, Xuefen Wang, Hector Barajas-Martinez.

**Software:** Ganxiao Chen, Xuefen Wang, Hector Barajas-Martinez, Dan Hu.

**Supervision:** Ganxiao Chen, Zuojiong Gong, Hector Barajas-Martinez, Dan Hu.

**Validation:** Ganxiao Chen, Zuojiong Gong, Hao Xia, Hector Barajas-Martinez.

**Visualization:** Ganxiao Chen, Zuojiong Gong, Yao Wang, Dan Hu.

**Writing – original draft:** Ganxiao Chen, Xun Li, Zuojiong Gong, Yao Wang, Yan Huang, Dan Hu.

**Writing – review & editing:** Ganxiao Chen, Xun Li, Zuojiong Gong, Hao Xia, Dan Hu.

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
