## [Decision Letter · Decision Letter 0]

17 Mar 2021

PONE-D-21-02635

Hypertension as a sequela in patients of SARS-CoV-2 infection

PLOS ONE

Dear Dr. Hu,

Thank you for submitting your manuscript to PLOS ONE. After careful consideration, we feel that it has merit but does not fully meet PLOS ONE’s publication criteria as it currently stands. Therefore, we invite you to submit a revised version of the manuscript that addresses the points raised during the review process.

I reviewed the paper along with reviewer comments  your paper will need more  data to support  your hypothesis  and major  revisions along with changes in grammar  and language, before it is considered for publication. In the current existing format and available data mentioned in the paper . it is difficult to establish correlation. Please revise your paper before it is considered for  review for publication .

We look forward to receiving your revised manuscript.

Kind regards,

Bhagwan Dass, MD

Academic Editor

PLOS ONE

Journal Requirements:

4. Thank you for stating the following in the Funding Section of your manuscript:

"The current work was supported by the National Natural Science Foundation

Project of China (Grant No. 81670304 – D.H.)."

"No - The funders had no role in study design, data collection and analysis, decision to publish, or preparation of the manuscript."

Reviewers' comments:

Reviewer's Responses to Questions

**Comments to the Author**

1. Is the manuscript technically sound, and do the data support the conclusions?

Reviewer #1: No

Reviewer #2: Partly

Reviewer #3: Partly

2. Has the statistical analysis been performed appropriately and rigorously? 

Reviewer #1: I Don't Know

Reviewer #2: Yes

Reviewer #3: Yes

3. Have the authors made all data underlying the findings in their manuscript fully available?

Reviewer #1: Yes

Reviewer #2: Yes

Reviewer #3: Yes

4. Is the manuscript presented in an intelligible fashion and written in standard English?

Reviewer #1: Yes

Reviewer #2: No

Reviewer #3: Yes

5. Review Comments to the Author

Reviewer #1: The article is written in a clear manner, however it is unclear at what points the authors use parametric statistics and when they use non-parametric statistics- in this regard they offer no justification for their respective uses. There are only a few minor grammatical errors and only one instance where there is a reference to a paper but no reference number or citation was supplied in the main test. Most importantly, the authors suggest that cardiac injury results from secondary bacterial injury, which based on the design of the trial and that no supporting evidence is provided to this effect (ie any form of analysis of culture data), this represents a jump in logic that I do not believe is justifiable. The authors appear to make this supposition based on elevated procalcitonin levels alone- there are published studies that have shown a small subset of patients with severe to critical illness will actually have elevated procalcitonin levels without secondary bacterial co-infection. The confounding effect notwithstanding, this represents a conclusion which is not substantiated by the trial presented here. Moreover, given the confounding effect of hydroxychloroquine, which was the practice at the time of the study, the authors should make note of whether or not HCQ was used in any of these patients, which may actually provide an additional, interesting variable for their analysis. In the section for processes of patient screening, an appendix would be helpful to list what "chronic heart diseases" they considered for exclusion. In the results section, the wording is at times confusing, though technically correct- further revision of this section may help in further iterations of the submission, but do not in themselves constitute grounds for rejection. The very last sentence of the discussion section is too strong of an assertion, perhaps a softer wording of the assertion would be better received. Lastly, the study limitations section does a poor job of discussing the study limitations- heavy revision is advised here- in fact, this section does not discuss any limitations at all and appears to be only a continuation of either the discussion section or the introduction of the conclusions section.

Reviewer #2: The study was well planned, conducted, analyzed, and presented. The title is captivating and engaging. The introduction is appropriate and discussion is well written and thorough. The subject matter and idea is novel and well thought-out. The clinical point is very interesting and relevant and the paper should be published pending revisions.

The English and syntax of the paper needs a lot of work. The conclusion that bacterial superinfection may be a cause of the cardiac damage needs further evidence from the literature. Please, see attached file.

Reviewer #3: This manuscript seeks to address hypertension as a sequalae of COVID-19 disease. Though the clinical association is important, especially as we see more long haul COVID (PASC) in the world population, the manuscript attempts to relate multiple causation for hypertension, including secondary bacterial infection and mycocardial injury during concurrent SARS-CoV-2 infection. The role of ACE2 and the RAS symptom and their relationship to SARS-CoV-2 infectivity and also their role in hypertension would be a more relevant topic to discuss in this manuscript.

Major issues for this manuscript include disorganized data that appears to suggest causation of multiple factors, independently, related to COVID-19 disease severity. It is suggested that the manuscript focus on direct hypertension risk factors associated with the SARS-CoV-2 infection and not broad measurements of cTnl and procalcitonin levels we may be unrelated to the development of chronic hypertension.

Minor issues of this manuscript include various typographical and grammatical errors. There is also need to expand definitions of mild and moderate severity of disease as well. Please include the National Health Commission of China clinical severity scale and criteria to clarify this in the results section.

6. PLOS authors have the option to publish the peer review history of their article (what does this mean?). If published, this will include your full peer review and any attached files.

Reviewer #1: No

Reviewer #2: No

Reviewer #3: No

---

## [Author Response · Author response to Decision Letter 0]

10 Apr 2021

Response to Reviewers 

Hypertension as a sequela in patients of SARS-CoV-2 infection (PONE-D-21-02635)

We are grateful to reviewers and editors for the comments and have made every effort to modify the manuscript accordingly or to address the concerns. Table 1-7 are replaced with the modified ones. Please see our specific responses below.

Journal Requirements:

Answer: Thanks for this suggestion sincerely. The manuscript has been revised according to the PLOS ONE's style requirements.

2. We suggest you thoroughly copyedit your manuscript for language usage, spelling, and grammar. If you do not know anyone who can help you do this, you may wish to consider employing a professional scientific editing service. Whilst you may use any professional scientific editing service of your choice, PLOS has partnered with both American Journal Experts (AJE) and Editage to provide discounted services to PLOS authors. Both organizations have experience helping authors meet PLOS guidelines and can provide language editing, translation, manuscript formatting, and figure formatting to ensure your manuscript meets our submission guidelines. To take advantage of our partnership with AJE, visit the AJE website (http://learn.aje.com/plos/) for a 15% discount off AJE services. To take advantage of our partnership with Editage, visit the Editage website (www.editage.com) and enter referral code PLOSEDIT for a 15% discount off Editage services. If the PLOS editorial team finds any language issues in text that either AJE or Editage has edited, the service provider will re-edit the text for free.

Answer: We are grateful for the visionary suggestion. We have toned down this manuscript.

Answer: We greatly appreciate your constructive comment. We have included the ethics statement information in Line 115 - 116 Page 4 as “All participants provided written informed consent and agreed to use their medical record for research purpose.”

manuscript:

"The current work was supported by the National Natural Science Foundation

Project of China (Grant No. 81670304 – D.H.)."

"No - The funders had no role in study design, data collection and analysis, decision to publish, or preparation of the manuscript."

Answer: Thank you for your correction. We have removed any funding-related text from the manuscript and updated our Funding Statement section of the online submission form.

Answer: We re-make the Data Availability statement as suggested in Line 458 - 462 Page 15 as “Data cannot be shared publicly because of the contents including information that could compromise research participant privacy/consent. Data are available from the Renmin Hospital of Wuhan University Ethics Committee (contact via whdxrmyy@126.com) for researchers who meet the criteria for access to confidential data.”

 

Reviewer #1: 

1.The article is written in a clear manner, however it is unclear at what points the authors use parametric statistics and when they use non-parametric statistics- in this regard they offer no justification for their respective uses. 

Answer: The reviewer’s comment is well taken. We have added different symbols represent different test methods in Tables 1 - 7.

2.There are only a few minor grammatical errors and only one instance where there is a reference to a paper but no reference number or citation was supplied in the main test. 

Answer: We appreciate the reviewer’s valuable comments. The manuscript has been checked thoroughly again and some grammar errors were corrected. Besides, the reference number has been rearranged in some parts of the manuscript.

3.Most importantly, the authors suggest that cardiac injury results from secondary bacterial injury, which based on the design of the trial and that no supporting evidence is provided to this effect (ie any form of analysis of culture data), this represents a jump in logic that I do not believe is justifiable. The authors appear to make this supposition based on elevated procalcitonin levels alone- there are published studies that have shown a small subset of patients with severe to critical illness will actually have elevated procalcitonin levels without secondary bacterial co-infection. The confounding effect notwithstanding, this represents a conclusion which is not substantiated by the trial presented here. 

Answer: Thank you a lot for your important comment again. We agree with the reviewer and have revised the manuscript accordingly as follows.

First, we agree that it is not rigorous enough to draw the conclusion that cardiac injury results from secondary bacterial injury and decide to remove it. we have adjusted the discussion about cardiac injury in Line 245 - 252 Page 11 as “In this study, older patients with diabetes are more likely to suffer from cardiac injury. Our further analysis shows that the level of white blood cells, neutrophils, procalcitonin, C-reactive protein, lactate, and lactic dehydrogenase were positively associated with cardiac injury. Besides, cardiac injury occurred mostly in severe patients. Consequently, we hypothesize that severe type of COVID-19 characterized by acute inflammation response might be more prone to cardiac injury, especially in patients with preexisting cardiovascular disease. ” 

Besides, according to your suggestion, we have rearranged our data of renin-angiotensin system (RAS). These results, integrated into Table. 5A&B, suggesting that Ang Ⅱ levels were significantly higher in the elevated blood pressure group. Thus, we discuss in Line 300-308 Page13 as “On the other hand, sixteen patients with elevated blood pressure show significantly higher levels of cTnI than those normal blood pressure patients. Several studies have demonstrated Ang II direct or indirect effects on cardiomyocytes, some of which were related to pro-inflammatory and pro-hypertrophic responses. Especially when the balance between the ACE and ACE2 was disrupted in COVID-19 patients, the increase in Ang II actions could lead to myocardial inflammation, oxidative stress, and myocyte apoptosis. This hypothesis explains why elevated blood pressure could occur in parallel with mild cardiac injury of COVID-19 patients. ” 

Finally, we resummarize our conclusions in the Abstract part (Line 57 - 60 Page 2) as “Hypertension, sometimes accompanied by elevated cTnI, may occur in COVID-19 patients and become a sequela. Enhancing Ang II signaling, driven by SARS-CoV-2 infection, might play an important role in renin-angiotensin system and consequently lead to the development of hypertension in COVID-19”

4.Moreover, given the confounding effect of hydroxychloroquine, which was the practice at the time of the study, the authors should make note of whether or not HCQ was used in any of these patients, which may actually provide an additional, interesting variable for their analysis. 

Thanks for this suggestion sincerely. We have checked carefully and rearranged each patients’ clinical and self-reported data in this study, and all subjects had no medical history of hydroxychloroquine.

5.In the section for processes of patient screening, an appendix would be helpful to list what "chronic heart diseases" they considered for exclusion. In the results section, the wording is at times confusing, though technically correct- further revision of this section may help in further iterations of the submission, but do not in themselves constitute grounds for rejection. 

Answer: We are grateful for the visionary suggestion. We have redefined "chronic heart diseases" in the Method part (Line 119 - 120 Page 5) and Figure legends part (Line 452-453 Page 21), as “Chronic heart disease includes ischemic heart disease, arrhythmia, valvular disease, and heart failure.”

6.The very last sentence of the discussion section is too strong of an assertion, perhaps a softer wording of the assertion would be better received. Lastly, the study limitations section does a poor job of discussing the study limitations- heavy revision is advised here- in fact, this section does not discuss any limitations at all and appears to be only a continuation of either the discussion section or the introduction of the conclusions section.

Answer: Thanks for this suggestion sincerely. We have adjusted the last sentence of the discussion section in Line 300 - 308 Page13, as “On the other hand, sixteen patients with elevated blood pressure show significantly higher levels of cTnI than those normal blood pressure patients. Several studies have demonstrated Ang II direct or indirect effects on cardiomyocytes, some of which were related to pro-inflammatory and pro-hypertrophic responses. Especially when the balance between the ACE and ACE2 was disrupted in COVID-19 patients, the increase in Ang II actions could lead to myocardial inflammation, oxidative stress, and myocyte apoptosis. This hypothesis explains why elevated blood pressure could occur in parallel with mild cardiac injury of COVID-19 patients.

 We have re-written the part of study limitations in Line 309 - 320 Page 14 as “In the present study, we propose that hypertension is probably a sequela of SARS-CoV-2 infection. Although several studies of COVID-19 have been reported, there are few reports about the sequela of the disease likely due to lack of long-term clinical follow-up, which also applies to our present research. Next, it is difficult to analyze whether the blood pressure of COVID-19 patients with preexisting hypertension is further increased. Consequently, many patients could not be incorporated in the analysis because of history of hypertension, which results in a relatively low sample size. Besides, the present study uncovered rising Ang II as one possible mechanism that might result in hypertension in COVID-19. However, due to a lack of detection about ACE2 levels and other components, therefore, we cannot gain a comprehensive view of the virus-induced imbalance of RAS pathway. ”

 

Reviewer #2: The study was well planned, conducted, analyzed, and presented. The title is captivating and engaging. The introduction is appropriate and discussion is well written and thorough. The subject matter and idea is novel and well thought-out. The clinical point is very interesting and relevant and the paper should be published pending revisions.

The English and syntax of the paper needs a lot of work. The conclusion that bacterial superinfection may be a cause of the cardiac damage needs further evidence from the literature. Please, see attached file.

Answer: We are grateful for the visionary suggestion. We agree with the reviewer and have revised the manuscript accordingly as follows.

First, we agree that it is not rigorous enough to draw the conclusion that cardiac injury results from secondary bacterial injury and decide to remove it. we have adjusted the discussion about cardiac injury in Line 245 - 252 Page11 as “In this study, older patients with diabetes are more likely to suffer from cardiac injury. Our further analysis shows that the level of white blood cells, neutrophils, procalcitonin, C-reactive protein, lactate, and lactic dehydrogenase were positively associated with cardiac injury. Besides, cardiac injury occurred mostly in severe patients. Consequently, we hypothesize that severe type of COVID-19 characterized by acute inflammation response might be more prone to cardiac injury, especially in patients with preexisting cardiovascular disease. ” 

Besides, according to your suggestion, we have rearranged our data of renin-angiotensin system (RAS). These results, integrated into Table. 5A&B, suggesting that Ang Ⅱ levels were significantly higher in the elevated blood pressure group. Thus, we discuss in Line 300 - 308 Page13 as “On the other hand, sixteen patients with elevated blood pressure show significantly higher levels of cTnI than those normal blood pressure patients. Several studies have demonstrated Ang II direct or indirect effects on cardiomyocytes, some of which were related to pro-inflammatory and pro-hypertrophic responses. Especially when the balance between the ACE and ACE2 was disrupted in COVID-19 patients, the increase in Ang II actions could lead to myocardial inflammation, oxidative stress, and myocyte apoptosis. This hypothesis explains why elevated blood pressure could occur in parallel with mild cardiac injury of COVID-19 patients. ” 

Finally, we resummarize our conclusions in the Abstract part (Line 57 - 60 Page 2) as “Hypertension, sometimes accompanied by elevated cTnI, may occur in COVID-19 patients and become a sequela. Enhancing Ang II signaling, driven by SARS-CoV-2 infection, might play an important role in renin-angiotensin system and consequently lead to the development of hypertension in COVID-19”

 

Reviewer #3: This manuscript seeks to address hypertension as a sequalae of COVID-19 disease. Though the clinical association is important, especially as we see more long haul COVID (PASC) in the world population, the manuscript attempts to relate multiple causation for hypertension, including secondary bacterial infection and mycocardial injury during concurrent SARS-CoV-2 infection. The role of ACE2 and the RAS symptom and their relationship to SARS-CoV-2 infectivity and also their role in hypertension would be a more relevant topic to discuss in this manuscript.

Major issues for this manuscript include disorganized data that appears to suggest causation of multiple factors, independently, related to COVID-19 disease severity. It is suggested that the manuscript focus on direct hypertension risk factors associated with the SARS-CoV-2 infection and not broad measurements of cTnl and procalcitonin levels may be unrelated to the development of chronic hypertension.

Answer: We greatly appreciate your constructive comment. According to your suggestion, we have rearranged our data of renin-angiotensin system (RAS). These results, integrated into Table. 5A&B, suggesting that the increase in Ang II actions may direct or indirect effects on cardiomyocytes. 

Thus, we discuss in Line 300 - 308 Page13 as “On the other hand, sixteen patients with elevated blood pressure show significantly higher levels of cTnI than those normal blood pressure patients. Several studies have demonstrated Ang II direct or indirect effects on cardiomyocytes, some of which were related to pro-inflammatory and pro-hypertrophic responses. Especially when the balance between the ACE and ACE2 was disrupted in COVID-19 patients, the increase in Ang II actions could lead to myocardial inflammation, oxidative stress, and myocyte apoptosis. This hypothesis explains why elevated blood pressure could occur in parallel with mild cardiac injury of COVID-19 patients. ” 

Besides, we resummarize our conclusions in the Abstract part (Line 57 - 60 Page 2) as “Hypertension, sometimes accompanied by elevated cTnI, may occur in COVID-19 patients and become a sequela. Enhancing Ang II signaling, driven by SARS-CoV-2 infection, might play an important role in renin-angiotensin system and consequently lead to the development of hypertension in COVID-19”

Minor issues of this manuscript include various typographical and grammatical errors. There is also need to expand definitions of mild and moderate severity of disease as well. Please include the National Health Commission of China clinical severity scale and criteria to clarify this in the results section.

Answer: Thank you for your correction. According to your suggestion, we have defined mild and moderate severity of disease in the Method part (Line 90 - 93 Page 4) as “Mild type is defined as mild clinical symptoms and no pneumonia manifestation found in imaging. Moderate cases refer to those who present with fever and respiratory tract symptoms, etc. And have pneumonia manifestations found in imaging.”

Besides, in the Result part (Line 207 Page 9), the division criteria have been supplemented, as “Results of demographic and laboratory findings between the severe and non-severe group, based on guidelines of the National Health Commission of China, are shown in Table 6.”

---

## [Editor Report · Decision Letter 1]

15 Apr 2021

Hypertension as a sequela in patients of SARS-CoV-2 infection

PONE-D-21-02635R1

Dear Dr. Hu,

We’re pleased to inform you that your manuscript has been judged scientifically suitable for publication and will be formally accepted for publication once it meets all outstanding technical requirements.

Kind regards,

Bhagwan Dass, MD

Academic Editor

PLOS ONE
---

## [Editor Report · Acceptance letter]

19 Apr 2021

PONE-D-21-02635R1 

Hypertension as a sequela in patients of SARS-CoV-2 infection 

Dear Dr. Hu:

I'm pleased to inform you that your manuscript has been deemed suitable for publication in PLOS ONE. Congratulations! Your manuscript is now with our production department. 

Kind regards, 

on behalf of

Dr. Bhagwan Dass 

Academic Editor

PLOS ONE